# Protective Effects of a Lutein Ester Prodrug, Lutein Diglutaric Acid, against H_2_O_2_-Induced Oxidative Stress in Human Retinal Pigment Epithelial Cells

**DOI:** 10.3390/ijms22094722

**Published:** 2021-04-29

**Authors:** Chawanphat Muangnoi, Rianthong Phumsuay, Nattapong Jongjitphisut, Pasin Waikasikorn, Monsin Sangsawat, Paitoon Rashatasakhon, Luminita Paraoan, Pornchai Rojsitthisak

**Affiliations:** 1Cell and Animal Model Unit, Institute of Nutrition, Mahidol University, Nakhon Pathom 73170, Thailand; chawanphat.mua@mahidol.ac.th (C.M.); Rianthong_p@hotmail.com (R.P.); 2Natural Products for Ageing and Chronic Diseases Research Unit, Faculty of Pharmaceutical Sciences, Chulalongkorn University, Bangkok 10330, Thailand; nattapong.pharmcu@gmail.com (N.J.); stamppasin@gmail.com (P.W.); jansee_93@hotmail.com (M.S.); 3Pharmaceutical Sciences and Technology Program, Faculty of Pharmaceutical Sciences, Chulalongkorn University, Bangkok 10330, Thailand; 4Department of Chemistry, Faculty of Science, Chulalongkorn University, Bangkok 10330, Thailand; paitoon.r@chula.ac.th; 5Department of Eye and Vision Science, Institute of Life Course and Medical Sciences, University of Liverpool, Liverpool L7 8TX, UK; Luminita.Paraoan@liverpool.ac.uk; 6Department of Food and Pharmaceutical Chemistry, Faculty of Pharmaceutical Sciences, Chulalongkorn University, Bangkok 10330, Thailand

**Keywords:** age-related macular degeneration, human retinal pigmented epithelium, oxidative stress, lutein, lutein diglutaric acid

## Abstract

Oxidative stress-induced cell damage and death of the retinal pigmented epithelium (RPE), a polarized monolayer that maintains retinal health and homeostasis, lead to the development of age-related macular degeneration (AMD). Several studies show that the naturally occurring antioxidant Lutein (Lut) can protect RPE cells from oxidative stress. However, the poor solubility and low oral bioavailability limit the potential of Lut as a therapeutic agent. In this study, lutein diglutaric acid (Lut-DG), a prodrug of Lut, was synthesized and its ability to protect human ARPE-19 cells from oxidative stress was tested compared to Lut. Both Lut and Lut-DG significantly decreased H_2_O_2_-induced reactive oxygen species (ROS) production and protected RPE cells from oxidative stress-induced death. Moreover, the immunoblotting analysis indicated that both drugs exerted their protective effects by modulating phosphorylated MAPKs (p38, ERK1/2 and SAPK/JNK) and downstream molecules Bax, Bcl-2 and Cytochrome c. In addition, the enzymatic antioxidants glutathione peroxidase (GPx) and catalase (CAT) and non-enzymatic antioxidant glutathione (GSH) were enhanced in cells treated with Lut and Lut-DG. In all cases, Lut-DG was more effective than its parent drug against oxidative stress-induced damage to RPE cells. These findings highlight Lut-DG as a more potent compound than Lut with the protective effects against oxidative stress in RPE cells through the modulation of key MAPKs, apoptotic and antioxidant molecular pathways.

## 1. Introduction

Age-related macular degeneration (AMD) is a retina disease affecting the macular area, which is a cause of impaired vision and irreversible blindness in the elderly population in the developing world [1,2]. The prevalence of AMD is rising notably and its prevalence has been estimated to 288 million in 2040 because of exponential population aging [3,4]. AMD is characterized by loss of central and sharp vision due to the progressive degeneration of the macula [5]. There are two forms of AMD: the atrophic or dry AMD and the neovascular or wet AMD. The dry AMD is characterized by progressive dysfunction of the retinal pigment epithelium (RPE) leading to loss of photoreceptors and retinal degeneration, while the wet AMD is characterized by detachment of RPE accompanied by choroidal neovascularization with intraretinal or subretinal leakage and hemorrhage [6,7,8]. Even though AMD is a major health problem, the treatments that exist for AMD are limited and restricted to the wet form, with no current treatment existing for the dry form of AMD. Thus, the discovery and development of effective treatments are critically needed to prevent or delay AMD progression. Although the pathogenesis of AMD is not completely understood, dysfunction of the RPE plays a major role in the AMD progression and is an important feature of AMD pathogenesis [9]. The RPE, located between the Bruch’s Membrane (BM)/choroid and the photoreceptors, performs important functions that promote and maintain retinal health, including visual cycle involvement, digestion of spent photoreceptor outer segments (POS), and establishment of the blood-retinal barrier (BRB) [9,10].

Over the last decade, a growing body of studies proved that oxidative stress in RPE cells plays a crucial role in cell dysfunction and death, which leads to AMD progression and development [11,12,13]. Oxidative stress is characterized by the increase of reactive oxygen species (ROS), caused by an imbalance between production and accumulation of ROS in cells and tissues and the ability of a biological system to detoxify these reactive products [14]. ROS exert damaging effects on nucleic acids, proteins and lipids, which subsequently cause cellular dysfunction [11,15]. RPE cells have a high metabolic rate and exist in an environment that is abundant in endogenous ROS, such as O^−^, H_2_O_2_ and OH^−^. Long-term accumulation of oxidative damage leads to dysfunction of RPE cells and increases their susceptibility to oxidative stress. The production of ROS causes activation of signaling pathways, especially mitogen-activated protein kinases (MAPKs) pathway such as p38 mitogen-activated protein kinase (p38), stress-activated protein kinase/c-Jun N-terminal kinase (SAPK/JNK), and extracellular signal-regulated kinase (ERK). The activation of MAPKs results in the apoptosis and proliferation of RPE cells via apoptosis-associated proteins in the mitochondria, namely, cytochrome C, Bax (pro-apoptotic) and Bcl-2 (anti-apoptotic) proteins [16,17,18]. Clearing of ROS from the cells is critical for their survival, particularly in post-mitotic cells such as the RPE, as an accumulation of damaged proteins is destructive for the cells [19]. Therefore, the protection of RPE cells from oxidative damage may be an effective therapeutic strategy against AMD development.

Under normal physiological conditions, the collective effort of endogenous antioxidant defense mechanisms neutralizes cellular damage by ROS. RPE cells also contain a wide range of antioxidants such as glutathione (GSH), glutathione peroxidase (GPx), catalase (CAT), superoxide dismutase (SOD), heme oxygenase-1 (HO-1), NADPH dehydrogenase quinone 1 (NQO1) as well as vitamins C and E, which scavenge and/or decompose ROS [12,20,21]. With aging, the ability of RPE cells to counteract or utilize these ROS diminishes resulting in oxidative stress [12,22]. Decreased activity of antioxidant enzymes is also observed in aging and AMD eyes [23].

Various studies have shown the protective effects of dietary antioxidants on reducing oxidative stress in relation to the risk of AMD development [24,25,26,27,28]. Lutein (Lut) (Figure 1A), a bioactive compound in the carotenoid group that is found at high levels in green leafy vegetables such as spinach, kale and broccoli [29], has several activities as antioxidant, anti-inflammation and anti-cancer [30,31,32,33]. Lutein (Lut) also is the primary dietary xanthophyll pigment responsible for macular pigment optical density in primates [29,34]. Despite its positive effects, one major limitation of the use of Lut as a therapeutic agent is its poor bioavailability [12,35].

The prodrug approach has been shown to enhance pharmacological properties by improving physico-chemical and biopharmaceutical properties such as aqueous solubility, stability, bioavailability and biological activities [36,37,38,39,40,41,42]. We have previously demonstrated that curcumin digluatric acid, an ester prodrug of curcumin conjugated with diglutaric acid, improved the biological activities of curcumin both in vitro and in vivo [41,42], suggesting that glutaric acid could serve as a promoiety of bioactive molecules. The conjugation of Lut with diglutaric acid via an ester bond can possibly increase the pharmacological and biological activities of Lut. Therefore, in this study, we synthesized a novel ester prodrug of Lut, namely lutein diglutaric acid (Lut-DG). Lut-DG and Lut were evaluated on their protective effect against oxidative stress induced by H_2_O_2_ in RPE cells. The underlying molecular mechanisms by which these drugs exert their effects were also explored.

## 2. Results

### 2.1. Evaluation of Lut and Lut-DG Effect on Cell Viability of ARPE-19 Cells

The morphology of ARPE-19 cells used in the cytotoxicity evaluation of Lut and Lut-DG is shown in Figure 2A. Cytotoxicity against ARPE-19 cells was determined before evaluating the protective effects of Lut and Lut-DG against oxidative stress to obtain an optimal concentration that did not affect the viability of ARPE-19 cells. Cells were treated with Lut and Lut-DG (1–5 µM) for 24 h. The results showed that concentrations of up to 1 µM for both compounds did not affect the cell viability of ARPE-19 cells (Figure 2B). Therefore, the highest concentration of Lut and Lut-DG at 1 µM was used for subsequent experiments to ensure effective and sustained compound activity throughout treatment.

### 2.2. Evaluation of H_2_O_2_ on Cell Viability of ARPE-19 Cells

We next sought to determine the concentration of H_2_O_2_ required to cause approximately 50% reduction in ARPE-19 cell viability. ARPE-19 cells were treated with various concentrations of H_2_O_2_ (100–500 µM) for 30, 60 and 120 min. The results showed that the treatment of H_2_O_2_ reduced cell viability and increased ROS production of ARPE-19 cells in a concentration and time-dependent manner (Figure 3A,B). The data showed that H_2_O_2_ treatment for 1 h at a concentration of 400 µM was sufficient for a 50% reduction in cell viability (Figure 3A). We therefore used the above-determined concentration and time point for H_2_O_2_ treatment.

### 2.3. Effect of Lut and Lut-DG against H_2_O_2_-Induced Oxidative Stress in ARPE-19 Cells

We further determined and compared the protective effects of Lut and Lut-DG on oxidative stress in ARPE-19 cells, induced with H_2_O_2_ at 400 µM for 1 h. As expected, the data demonstrated that the cell viability was reduced by about 50% (Figure 4A). Pre-treatment of cells with Lut and Lut-DG for 24 h significantly protected cells from oxidative stress-induced cell death and ROS production (Figure 4A,B). When comparing Lut to Lut-DG, pre-treatment with Lut-DG caused a significantly better protective effect on cell viability and ROS production than Lut in the oxidative stress-induced ARPE-19 cells. Inverted microscopic analysis (Figure 4C) supported the protective effect of Lut and Lut-DG against H_2_O_2_-induced oxidative damage in ARPE-19 cells. These findings highlight Lut-DG as a more effective agent against oxidative stress than Lut.

### 2.4. Effect of Lut and Lut-DG against H_2_O_2_-Induced Oxidative Stress via Modulation of the MAPKs Pathway

This part of the study aimed to understand and explain the molecular mechanisms that modulated the protective effects of both Lut and Lut-DG on oxidative stress in ARPE-19 cells. Previous studies have confirmed that the activation of MAPKs such as the phosphorylation of p38, ERK1/2 (p44/42) and SAPK/JNK is strongly associated with the promotion of H_2_O_2_-induced cell death by apoptosis [43]. We therefore examined if the same pathway modulated H_2_O_2_-induced cell death observed in the present study. Analysis of protein expression via immunoblotting showed that ARPE-19 cells treated with 400 µM of H_2_O_2_ for 1 h significantly increased the expression of the phosphorylated forms of p38 (p-p38), ERK1/2 (p-ERK1/2) and SAPK/JNK (p-SAPK/JNK) compared with the control group (*p* < 0.05) (Figure 5A–C). These results indicate that H_2_O_2-_induced cell death occurs through the p-p38, p-p44/42 and p-SAPK/JNK pathways. Pre-treatment of ARPE-19 cells with Lut and Lut-DG for 24 h significantly decreased the expression of p-p38, p-ERK1/2 and p-SAPK/JNK compared to the H_2_O_2_-induced ARPE-19 cells group (*p* < 0.05). Lut-DG showed a more significant decrease in expression of p-p38, p-pERK1/2 and p-SAPK/JNK than Lut (Figure 5A–C), showing that Lut-DG is more effective than Lut in its ability to reverse oxidative stress-induced response.

### 2.5. Lut and Lut-DG Inhibit Apoptosis by Modulation of Bax, Bcl-2 and Cytochrome c Expression in Oxidative Stressed ARPE-19 Cells

To understand the molecular mechanisms of Lut and Lut-DG in providing a protective effect against oxidative stress, we investigated the apoptosis pathway by determining the expression levels of molecules downstream of the MAPKs signaling pathway, specifically Bax (pro-apoptotic), Bcl-2 (anti-apoptotic) and Cytochrome c proteins [44,45]. The protein expression of Bax, Bcl-2 and Cytochrome c were determined by immunoblotting in ARPE-19 cell cultures (Figure 6A–C). Analysis of protein expression showed that H_2_O_2_ treatment significantly and simultaneously increased Bax and Cytochrome c, and decreased Bcl-2 levels in ARPE-19 cultures compared to the control group. In ARPE-19 cells treated with H_2_O_2_, pre-incubation with Lut and Lut-DG for 24 h led to a significant decrease in Bax and Cytochrome c protein levels and a significant increase in Bcl-2 protein levels compared to the control (Figure 6A–C). Therefore, Lut and Lut-DG exert their protective effect against oxidative stress-induced cell death via modulation of the Bax, Bcl-2 and Cytochrome c in the apoptosis pathway. Lut-DG was a more effective agent in its protective effect against oxidative stress as shown by its enhanced ability to reverse the oxidative-induced alteration of Bax, Bcl-2 and Cytochrome c protein expression compared to Lut treatment.

### 2.6. Lut and Lut-DG Exert Their Protective Effect against Oxidative Stress-Induced Cell Death via Modulation of Key Non-Enzymatic and Enzymatic Antioxidants in ARPE-19 Cells

Enzymatic antioxidants such as catalase (CAT) and glutathione peroxidase (GPx), and non-enzymatic antioxidants, such as glutathione (GSH), can protect against oxidative damage by counteracting high levels of ROS in cells [46]. Therefore, we examined whether Lut and Lut-DG modulate GPx and CAT activities and GSH level to protect ARPE-19 cells against oxidative stress.

CAT and GPx activities and GSH level showed that respective H_2_O_2_ treatment significantly decreased activity and level of these antioxidants compared to the control group without any treatment (Figure 7A–C). However, pre-treatment with Lut and Lut-DG significantly enhanced CAT and GPx activities and GSH level in ARPE-19 cells subjected to oxidative stress (Figure 7A–C). Interestingly, we observed that Lut treatment only significantly increased CAT activity compared to the control cells, whereas Lut-DG significantly increased CAT and GPx activities and a GSH level compared to the control cells (Figure 7A–C). The results suggest that pre-treatment of Lut and Lut-DG can improve the antioxidant system of the RPE in order to be armed against potential oxidative stress inducers. In all cases, Lut-DG provided a more effective response in enzymatic and non-enzymatic antioxidant systems compared to Lut.

## 3. Discussion

The present study demonstrated, for the first time to our knowledge, the protective effects of Lut-DG, an ester prodrug form of Lut, against oxidative stress in human RPE cells. At the molecular level, we showed that both Lut and Lut-DG exert their protective effects through modulation of the key apoptotic-signaling pathway p38, ERK1/2 (p44/42) and SAPK/JNK and its downstream effector molecules Bax, Bcl-2 and Cytochrome c. In addition, Lut and Lut-DG were also able to protect against oxidative damage by increasing the expression of key enzymatic (CAT and GPx) and non-enzymatic (GSH) antioxidant systems in RPE cells. Furthermore, we showed that Lut-DG gives a better protective effect than Lut against oxidative insult in RPE cells and highlights the potential use of Lut-DG as an alternative therapeutic agent for AMD, a disease significantly linked to oxidative-induced RPE dysfunction and cell death.

H_2_O_2_ was chosen to induce oxidative stress in ARPE-19 cells because ARPE-19 cells have a high metabolic rate and exist in an environment that is abundant in endogenous ROS, such as O2-, H_2_O_2_ and OH-. Long-term accumulation of oxidative damage leads to dysfunction of RPE cells and increases their susceptibility to oxidative stress [47,48]. H_2_O_2_ is one of the most common oxidants used in the cellular oxidative stress models for ARPE-19 cells. [49,50,51]. The increase of the intracellular H_2_O_2_ level in response to various pro-oxidants can further induce excessive ROS production in the cells which leads to the RPE cells dysfunction and death from oxidative.

We compared the protective effects of Lut and Lut-DG against oxidative stress in RPE cells. Lut has been shown to be protective against oxidative stress for eye health in different cell models such as in human lens epithelial cells [52] and retinal ganglion cells [53]. As mentioned previously, Lut also protects against oxidative-induced RPE dysfunction and cell death [28,54,55]. Therefore, it was hypothesized that the prodrug of Lut, Lut-DG, would exert a similar effect against oxidative stress. As expected, we observed that the ARPE-19 cells pre-treated with Lut and Lut-DG alleviated H_2_O_2-_induced ROS production and improved cell viability. More importantly, Lut-DG exerted a more potent protective effect against oxidative stress, as shown by a lower ROS production and a higher level of viable cells than Lut-treated cells. The addition of glutaric acid to Lut does not affect the length of the conjugating system, suggesting that the increase in ROS-protective action of Lut-DG does not result from an increase of the conjugating system. Ester prodrugs of Lut with various pro-moieties such as palmitate, myristate, linoleate or laureate were more stable than Lut under storage at 25 °C in the dark [56]. In addition, lutein dimyristate was more stable than Lut against heat and UV light conditions than Lut [57]. Therefore, it is likely that the elevated ROS-protective action of Lut-DG compared to Lut may result from a longer hydrocarbon chain, which better stabilizes the delocalized electron cloud of the entire Lut-DG compound.

Several studies showed that the prodrug and formulation development of drugs and bioactive compounds could improve bioavailability and biological activities [39,40,41,58]. A previous study showed that the phospholipid complex of quercetin could improve the solubility of quercetin and enhance the protective effect against oxidative-induced damages in ARPE-19 cells compared to quercetin [50]. A recent study showed that an ester prodrug of Cur, the so-called curcumin diethyl disuccinate, which was developed to increase the stability and bioavailability of curcumin, was approximately 3X more potent than Cur in protecting RPE cells from stress-induced injury [25]. Similarly, we showed Lut-DG is more effective and potent than Lut regarding the protective effects on oxidative-induced RPE injury. This increased potency of Lut-DG also showed that the prodrug form of Lut improved drug bioavailability.

We further determined that Lut and Lut-DG were able to exert their protective effect against oxidative stress through modulation of MAPKs, namely the p38 mitogen-activated protein kinase (p38), stress-activated protein kinase/c-Jun N-terminal kinase (SAPK/JNK) and extracellular signal-regulated kinase (ERK) signaling pathway. These signaling pathways are involved in key cellular functions such as cell proliferation and apoptosis. In general, transient (less than 15 min) activation of this pathway is important for cell proliferation and survival [59], whereas sustained activation of the p38, ERK and SAPK/JNK can induce cell death [60]. Activation of these MAPKs is able to influence the expression of downstream apoptotic regulators Bax (pro-apoptotic), Bcl-2 (anti-apoptotic) and Cytochrome c [61,62]. The balance between Bax and Bcl-2 determines whether apoptosis is triggered [63,64]. The activation of ERK1/2 is especially strongly associated with the promotion of H_2_O_2_-induced cell apoptosis in renal epithelial cells [43]. Furthermore, a recent study showed that Lut was able to inhibit apoptosis through modulating Bax/Bcl2 and Cytochrome c expression and reduced oxidative stress in testes of rat Muller and photoreceptor cells [65,66]. Thus, investigating the activation of MAPKs and expression of downstream molecules Bax, Bcl-2 and Cytochrome c allows an understanding of how H_2_O_2_ induces apoptosis in RPE cells. In this study, H_2_O_2_ treatment led to an increase in protein levels of the phosphorylation of p38 (p-p38), ERK1/2 (p-ERK1/2) and SAPK/JNK (p-SAPK/JNK) in ARPE-19 cells. The result was consistent with an increase in the protein level of Bax and Cytochrome c as well as a reduction in the protein level of Bcl-2. These findings showed that cell death observed in RPE cells upon H_2_O_2_ treatment was through the apoptotic pathway. Pre-treatment with Lut and Lut-DG prevented these changes leading to increased cell viability against oxidative stress. Notably, Lut-DG was more effective in modulating these apoptotic effectors than Lut in this RPE cell model.

In addition to apoptotic signaling pathways, we suggested that Lut and Lut-DG were able to influence enzymatic antioxidants such as glutathione peroxidase (GPx) and catalase (CAT), and non-enzymatic antioxidant glutathione (GSH). We demonstrated that cells pre-treated with Lut and Lut-DG presented increased activities of CAT and GPx, and a higher level of GSH, thus indicating that these compounds protected cells from oxidative stress induced by the H_2_O_2_ treatment. Moreover, Lut-DG exerted a more potent effect in modulating these enzymatic and non-enzymatic antioxidants than Lut. The influence of these drugs on the expression of antioxidant enzymes was expected as Lut, which is a potent scavenger of ROS, can function indirectly as an antioxidant by increasing the activities and the levels of these key enzymatic and non-enzymatic antioxidants [67,68]. In RPE cells, it has been previously reported that Lut protected RPE cells against oxidative stress by activating enzymatic and non-enzymatic antioxidant systems through transcription factor Nrf2 leading to reduced levels of ROS [54]. Another study showed that Lut administration increased enzymatic activity and the levels of non-enzymatic antioxidants in a rat model with induced oxidative stress and inflammation with lipopolysaccharide [32].

In the present study, cells under oxidative stress by incubating with H_2_O_2_ showed a decrease in the activities of CAT and GPx and the levels of GSH. Cells pre-treated with Lut or Lut-DG showed increased activities of CAT and GPx and the level of GSH compared to the oxidatively stressed cells. Moreover, we found that under normal (in the absence of oxidative stress) conditions, Lut treatment significantly increased CAT activity compared to control cells, and Lut-DG significantly increased activities of both CAT and GPx and the level of GSH compared to the control cells. These results indicate that Lut-DG treatment could support oxidative stress protection in RPE cells by enhancing both activities and the levels of enzymatic and non-enzymatic antioxidants. In addition, the increase of CAT and GPx activities and the GSH level were previously related to the activation of the transcription factor nuclear factor erythroid 2–related factor 2 (Nrf2) located inside the cells [50,51,69]. These results suggest that the Lut-DG can enter the cells. However, the possibility of Lut-DG and Lut localization on the surface of the cell using Lut-DG or Lut with radioactive or fluorescent labeling should be further investigated.

The signaling mechanisms that regulate the activities of CAT and GPx and the level of GSH were not investigated in this present study. It was previously reported that antioxidants protect oxidatively stressed RPE cells through activation of the Akt/Nrf2 signaling pathway. This pathway involves the translocation of Nrf2 into the nucleus, which results in the expression of several enzymatic and non-enzymatic antioxidants [70,71]. Studies have shown that Lut exerts its protective effects against oxidative stress by activating the Nrf2 signaling [32,54]. It is likely that Lut and Lut-DG can influence the levels of the enzymatic and non-enzymatic antioxidants through this pathway.

This study investigated the protective effect and mechanism of Lut-DG on H_2_O_2_-induced oxidative stress in ARPE-19 cells. The oxidative stress in ARPE-19 cells plays a crucial role in cell dysfunction and death, which leads to AMD progression and development. Our results indicate that Lut-DG is a promising prodrug of Lut and has the potential to be extensively investigated as a therapeutic agent or an adjuvant for the treatment of AMD in additional preclinical and clinical studies. However, RPE cells in vivo are mostly not able to divide due to steric constraints, but they can do so when in culture. Moreover, senescence of RPE, and not cell death, may underline retinal degeneration.

## 4. Materials and Methods

### 4.1. Chemicals and Reagents

Lut was purchased from Energy Chemical (Energy Chemical, Shanghai, China) and used as a starting material to synthesize Lut-DG. Lut-DG was synthesized and characterized by Infrared (IR) and Nuclear Magnetic Resonance (NMR) spectrometry as described below. All the solvents and reagents used for the synthesis were of analytical grade and purchased from commercial sources without further purification. Glutaric anhydride was purchased from TCI (Tokyo Chemical Industry, Tokyo, Japan). N, N-Diisopropylethylamine (DIPEA) was purchased from Oakwood chemical (Oakwood Products, South Carolina, USA). The 1H NMR spectrum was recorded on a Jeol 500 MHz spectrometer, using chloroform-D as a solvent. The IR spectrum was carried out on a Bruker/Alpha platinum ATR. The 2′,7′-dichlorofluorescein diacetate (DCFH-DA) was purchased from Sigma Aldrich (Sigma-Aldrich, Dorset, UK). Fetal bovine serum (FBS) was obtained from Merck Millipore (Merck Millipore, Darmstadt, Germany). Dulbecco’s modified Eagle’s Medium/Nutrient Mixture F-12 Ham (DMEM/F-12), penicillin-streptomycin, hydrogen peroxide and dimethyl sulfoxide (DMSO)3-[4,5-dimethyltiazol-2-yl]-2,5-diphenyl-tetrazolium bromide (MTT) were purchased from Invitrogen (Invitrogen Ltd., Paisley, UK). All antibodies for western blot were purchased from Cell Signaling Technology (Danvers, MA, USA). All kits for antioxidant enzyme assays were purchased from Cayman Chemical (Cayman Chemical, Ann Arbor, MI, USA).

### 4.2. Synthesis of Lutein Diglutaric Acid (Lut-DG)

N, N-diisopropylethylamine (DIPEA, 0.6 mL, 2.1 mmol) was added to a 10 mL solution of glutaric anhydride (240 mg, 2.1 mmol) in anhydrous dichloromethane. The mixture was stirred and gradually added with a solution of lutein (200 mg, 0.35 mmol) in anhydrous dichloromethane (5 mL). The reaction mixture was stirred under a nitrogen atmosphere and light protection at room temperature until the completion of the reaction (24 h) based on TLC examination (Figure 8). After the volatile solvents were evaporated under reduced pressure, the residue was redissolved with ethyl acetate, then washed with 0.1 M HCl and water. The organic phase was separated and extracted with 5% NaHCO_3_. The aqueous layer was collected and acidified by 3.0 M HCl. This solution was extracted again with dichloromethane. The combined organic layer was dried over anhydrous sodium sulfate, filtered, and concentrated under reduced pressure. The red powder of Lut-DG was obtained in a 36% yield. The purity of Lut-DG was over 95% analyzing by Agilent 1290 Infinity UHPLC system. The chemical structure of Lut-DG was characterized by IR: 2500–3500 cm^−1^ (broad, -COOH), 2919 and 2851 cm^−1^ (C-H stretching of -CH_2_ and -CH_3_), 1706 cm^−1^ (carbonyl of ester, C=O), 1145 cm^−1^ (C-O stretching), 963 cm^−1^ (alkene bending, C=C). ^1^H-NMR (500 MHz, CDCl_3_) δ 6.68–6.04 (m, 14H), 5.51–5.46 (m, 1H), 5.46–5.39 (m, 1H), 5.36–5.30 (m, 1H), 5.11–5.03 (m, 1H), 2.49–2.36 (m, 10H), 2.05 (s, 1H), 2.01–1.93 (m, 16H), 1.87–1.81 (m, 1H), 1.72 (s, 4H), 1.65 (s, 3H), 1.47–1.41 (m, 1H), 1.40–1.36 (m, 1H), 1.09 (m, 9H), 0.87 (s, 3H).

### 4.3. Cell Culture of ARPE-19 Cells

An authenticated cell line ARPE-19 (ATCC, Rockville, MD, USA) was used as a model to represent human RPE cells. These cells were routinely maintained and cultured in a 1:1 mixture of DMEM/F-12 supplemented with 10% (*v*/*v*) heat-inactivated FBS and 1% (*v*/*v*) pen-strep. All experiments were carried out in standard 6-well or 24-well plates unless stated otherwise. For the 6-well plate set-up, ARPE-19 cells were seeded at a cell density of 1.0 × 10^6^ cells/well. For a 24-well plate set-up, ARPE-19 cells were seeded at a cell density of 0.2 × 10^6^ cells/well. Cells were then grown to obtain a confluent monolayer and then used for subsequent experiments.

### 4.4. Cell Viability (MTT) Assay

Cell viability was measured using the 3-(4, 5-dimethlthiazol-2-yl)-2, 5-diphenyltetrazolium bromide tetrazolium (MTT) assay. The MTT substrate is converted into a purple-colored formazan product from incubation with cells, which is proportional to the number of live cells. After cells reached their respective experimental time points, media was removed, and cells were washed one time with phosphate buffer saline (PBS) and incubated with PBS containing 0.5 mg/mL of MTT at 37 °C for 4 h. After incubation, the MTT solution was removed from the cells and DMSO was added to solubilize formazan crystal produced from MTT by viable cells. Cells were incubated at 37 °C for 10 min, after which absorbance was measured at 540 nm using a microplate reader (SPECTROstar^Nano^, BMG LABTECH, Germany).

### 4.5. Evaluation of Cytotoxicity of Lut and Lut-DG

Media was removed from confluent ARPE-19 cultures in a 24-well plate set up and, after washing with excess serum-free media, cells were incubated with different Lut or Lut-DG concentrations (0.1, 0.5, 1, 2 and 5 µM) for 24 h. A 0.5% DMSO vehicle control well was included in the experiment. After the respective experimental time, cell viability was measured by MTT assay.

### 4.6. Evaluation of Suitable H_2_O_2_ Concentration for Cytotoxicity Induction

Media was removed from confluent ARPE-19 cultures in a 24-well plate set up and, after washing with excess serum-free media, cells were incubated with different H_2_O_2_ concentrations (100, 200, 300, 400 and 500 µM) in a serum-free medium at 37 °C for 30, 60, and 120 min. Control cells were cultured in a serum-free medium without H_2_O_2_. After respective experimental treatment, cells were washed twice with excess PBS, after which cell viability was determined by MTT assay.

### 4.7. Evaluation of the Protective Effect of Lut- and Lut-DG-Induced Oxidative Stress on ARPE-19 Cells

ARPE-19 cells cultured in 6-well and 24-well plate set up were washed with serum-free media, after which appropriate wells were incubated for 24 h with either Lut and Lut-DG (both at a concentration of 1 µM) in serum-free media. A 0.5% DMSO vehicle control well was included in the experiment. After pre-treatment with Lut and Lut-DG, cells were washed with PBS and then incubated with appropriate H_2_O_2_ concentrations in serum-free medium at 37 °C for 1 h. The 24-well plate set up was used for cell viability measurement using the MTT assay. The 6-well plate set up was used for enzymatic and non-enzymatic antioxidant assays and western immunoblot analysis.

### 4.8. Evaluation of ROS Production

DCFH-DA was used to detect and quantify intracellular ROS activity. ARPE-19 cells were seeded into wells of a black 96-well, clear bottom plates (Corning, New York, USA) at a cell density of 3.0 × 10^4^ cells/well and cultured until confluence. Media was then removed, and cells were washed with excess PBS. After washing, appropriate wells were incubated for 24 h with either Lut or Lut-DG (both at a concentration of 1 µM) in serum-free media. A 0.5% DMSO vehicle control well was included in the experiment. After the respective experimental time point, media was removed and cells were washed with serum-free media. Next, cells were incubated with 10 µM DCFH-DA in serum-free media at 37 °C for 20 min. The cells were then washed with serum-free media followed by incubation with appropriate concentrations of H_2_O_2_ in serum-free media at 37 °C for 1 h. Finally, cells were washed twice with PBS, and ROS production was measured in the plates using the Fluostar Optima plate reader (BMG Labtech, Aylesbury, UK) with the excitation/emission settings of 485 nm/530 nm, respectively.

### 4.9. CAT, GPx and GSH Determination

ARPE-19 cells from a 6-well plate set up were used for the preparation of cellular lysates. Cells were scraped and incubated with PBS containing 0.5% (*v*/*v*) triton-x100. The cell solution was sonicated in an ultrasonic sonicator at 4 °C for 10 min. Cell lysates were centrifuged at 14,000× *g* at 4 °C for 10 min. The supernatant was collected to determine CAT and GPx activities, and GSH levels using the respective assay kits.

For CAT assays, cell lysate (20 μL) was gently mixed with 100 μL of assay buffer and 100 μL of methanol in each 96-well plate. To initiate the reaction, 20 μL of H_2_O_2_ solution (35 mM) was added to the lysate solution. The major end product of this reaction is formaldehyde. Hence, the rate of formaldehyde production from peroxidation reaction can be used to estimate CAT activity. After incubation on a shaker for 20 min, 30 μL of potassium hydroxide (10 M) was added to terminate the reaction. The sample was subsequently incubated with 30 μL of 4-amino-3-hydrazino-5-mercapto-1,2,4-triazole (purpaid) for 10 min. Purpaid is used as a chromogen to specifically interact with the formaldehyde generated from peroxidation reaction. The 10 μL of potassium periodate (0.5 M) was added into each well to complete the oxidation reaction and develop a purple-color product. After 5 min of incubation, the alteration in color (from colorless to purple) was observed at a wavelength of 540 nm using a SPECTROstar microplate reader. Standard curves with serial dilutions of formaldehyde solution (0–85 μM) were used in each experiment to estimate the amount of generated formaldehyde in the sample. The activity of CAT in the sample was further calculated as units per mg of protein.

For GPx assay, cell lysate (20 μL) was mixed with 50 μL of the co-substrate mixture (glutathione reductase and GSH), 50 μL of NADPH and 50 μL of assay buffer in each 96-well plate. To initiate the reaction, 20 μL of cumene hydroperoxide solution was added into each well. After adding cumene hydroperoxide, the absorbance at 340 nm was monitored every 1 min for 10 min with a SPECTROstar microplate reader. The GPx activity in lysate samples was then calculated from a rate of decrease in the absorbance of samples. The GPx activity was expressed as units per mg of protein.

For GSH assay, cell lysate (50 μL) was mixed with the 150 μL of the assay cocktail (GSH MES buffer, GSH co-factor mixture, GSH enzyme mixture and GSH DTNB). The plate was incubated in the dark on an orbital shaker. After 20 min of incubation, the absorbance was observed at a wavelength of 540 nm using a SPECTROstar microplate reader. Standard curves with serial dilutions of GSH (0–16 μM) were used in each experiment to estimate the amount of GSH in the sample. The level of GSG in the sample was further calculated as units per mg of protein.

### 4.10. Western Immunoblot Analysis

ARPE-19 cells from a 6-well plate set up were used for the preparation of cellular protein lysates. Cell protein lysates were prepared in lysis buffer [72] and subjected to immunoblotting. Once collected, cell lysates were centrifuged at 14,000× *g* at 4 °C for 10 min. The supernatants were retained and the protein concentrations were determined using the BCA protein assay kits (Thermo Scientific, Rockford, IL, USA). Equal amounts (40 µg) of protein samples were separated by 10% SDS-PAGE; the resolved proteins were then transferred to nitrocellulose membrane (Amersham™ Protran^®^, Sigma Aldrich), after which the membrane was blocked with 5% dry milk. The primary and secondary antibodies used are listed in Table 1. After primary and secondary antibody steps, protein detection was achieved using an enhanced chemiluminescent detection kit (Thermo Scientific, Rockford, IL, USA) followed by imaging on the BioRad ChemiDoc^TM^ (BioRad, Hampstead, UK). The results were analyzed by Image J software (BioRad, Hampstead, UK) to obtain the optical density. Band densitometry values of p-p38, p-ERK1/2, and p-SAPK/JNK were normalized to the band intensity of p38, ERK1/2, and SAPK/JNK, respectively, and densitometry values of Bax, Bcl-2 and Cytochrome c were normalized to the band of beta-actin.

### 4.11. Statistical Analysis

All data are presented as the mean values ± standard deviations (SD) of a minimum of three independent experiments. Data analysis was performed using the commercially available software SPSS (Version 16.0). A *p*-value of ≤ 0.05 was considered statistically significant.

## 5. Conclusions

In summary, the study evidences that the effect of Lut and Lut-DG on the apoptosis-related proteins is oxidative stress-dependent in the H_2_O_2_-induced oxidative stress in human ARPE-19 cells. The effect on the apoptosis-related proteins is oxidative stress-dependent in the H_2_O_2_-induced oxidative stress in human ARPE-19 cells. Therefore, the H_2_O_2_-induced oxidative stress in human ARPE-19 cells is applicable as a model for evaluating the ability of lutein and its prodrug, lutein diglutaric acid (Lut-DG), to protect these cells from oxidative stress. We demonstrate that Lut-DG is a more potent protective antioxidant than Lut against oxidative damage to RPE cells through suppression of ROS levels and protection from cell death. The molecular mechanisms by which Lut-DG showed its effects were through specific modulation of apoptotic signaling (p38, ERK1/2 and SAPK/JNK) and downstream proteins such as Bax, Bcl2 and Cytochrome c as well as enhancement of activities of enzymatic antioxidants CAT and GPx and the level of non-enzymatic antioxidant GSH. The results provide evidence for Lut-DG as a promising therapeutic agent for the treatment of AMD.

## Figures and Tables

**Figure 1 ijms-22-04722-f001:**
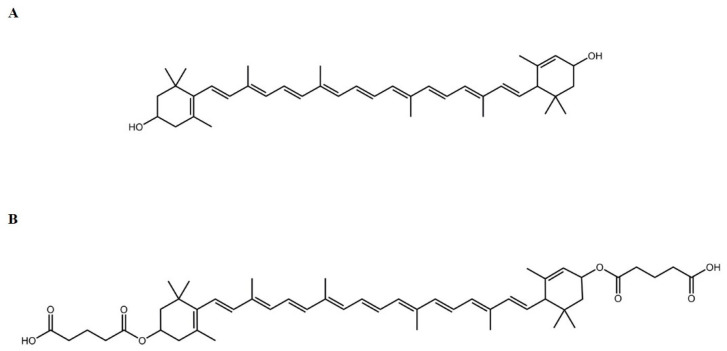
Structure of compounds used in the present study. (**A**) Parent drug Lut; (**B**) Prodrug lutein diglutaric acid (Lut-DG).

**Figure 2 ijms-22-04722-f002:**
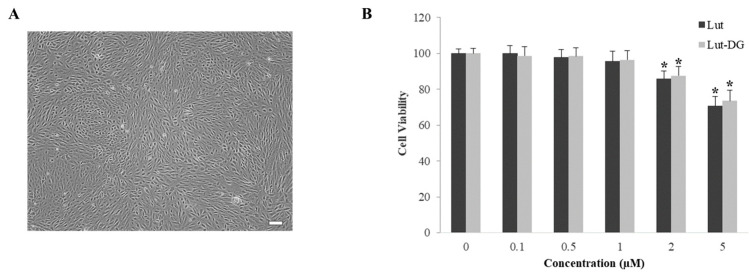
Effect of Lut and Lut-DG on cell viability of ARPE-19 cells. (**A**) Morphology by phase-contrast microscopy of ARPE-19 cells. Scale bar represents 100 µm; (**B**) ARPE-19 cells were treated with a concentration range (0.1–5 µM) of Lut and Lut-DG for 24 h, after which cell viability was measured using MTT assay. Graphs represent average cell viability (mean ± SD values, *n* = 4; (One-Way ANOVA test), * *p* ≤ 0.05 vs. control group).

**Figure 3 ijms-22-04722-f003:**
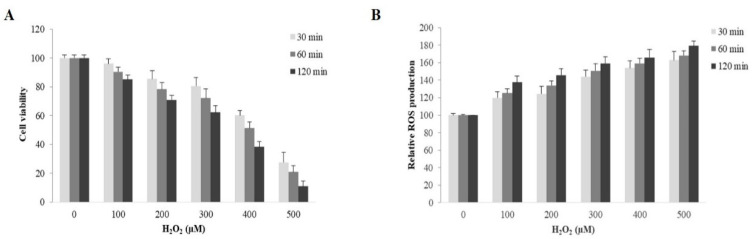
Evaluation of H_2_O_2_ concentration and exposure time needed for cell death induction and ROS generation in ARPE-19 cells. (**A**) Cell death induction and (**B**) reactive oxygen species (ROS) generation in ARPE-19 cells treated with various concentrations of H_2_O_2_ (100–500 µM) for 30, 60 and 120 min; cell viability and ROS production were measured using MTT and DCFH-DA assays, respectively. Graphs represent average cell viability and relative ROS production (mean ± SD values, *n* = 4).

**Figure 4 ijms-22-04722-f004:**
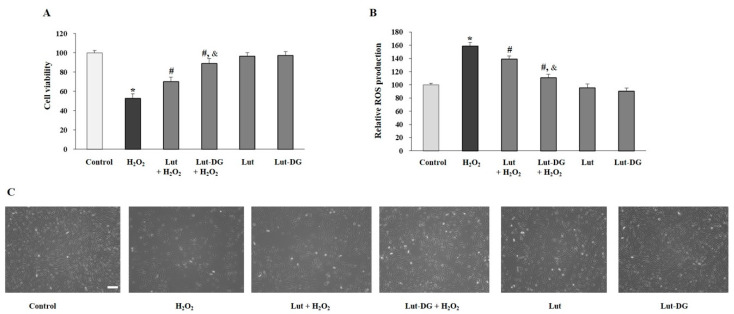
Protective effects of Lut and Lut-DG on H_2_O_2_-induced cytotoxicity and ROS production in ARPE-19 cells. (**A**) ARPE-19 cells were pre-treated with 1 µM of Lut and Lut-DG for 24 h, followed by H_2_O_2_ treatment at 400 µM 1 h. Cell viability was measured using MTT assay; (**B**) ROS generation was determined by DCFH-DA assay. Graphs represent average cell viability and ROS production (mean ± SD values, *n* = 4; (One-Way ANOVA test), * *p* ≤ 0.05 vs. control group, ^#^
*p* ≤ 0.05 vs. H_2_O_2_ group, and ^&^ *p* ≤ 0.05 vs. Lut + H_2_O_2_ group); (**C**) Morphology by phase-contrast microscopy of ARPE-19 cells under all experimental conditions. Scale bar represents 100 µm.

**Figure 5 ijms-22-04722-f005:**
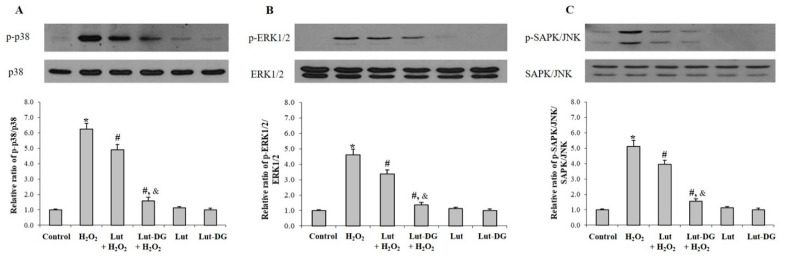
Protective effects of Lut and Lut-DG against oxidative stress occurs through modulation of the apoptotic MAPKs signaling pathway. ARPE-19 cells were pre-incubated with 1 µM of Lut and Lut-DG for 24 h, followed by H_2_O_2_ treatment at 400 µM for 1 h. Protein levels of phosphorylated (**A**) p38, (**B**) ERK1/2 and (**C**) SAPK/JNK were determined by immunoblotting of cell lysates. Band densitometry values of p-p38, p-ERK1/2 and p-SAPK/JNK were normalized to the band of p38, ERK1/2, and SAPK/JNK, respectively. Representative western blots are shown, with graphs presenting average normalized protein expression. Results are presented as mean ± SD values, *n* = 4; (One-Way ANOVA test), *****
*p* ≤ 0.05 vs. control group, **^#^**
*p* ≤ 0.05 vs. H_2_O_2_ group, and ^&^ *p* ≤ 0.05 vs. Lut + H_2_O_2_ group).

**Figure 6 ijms-22-04722-f006:**
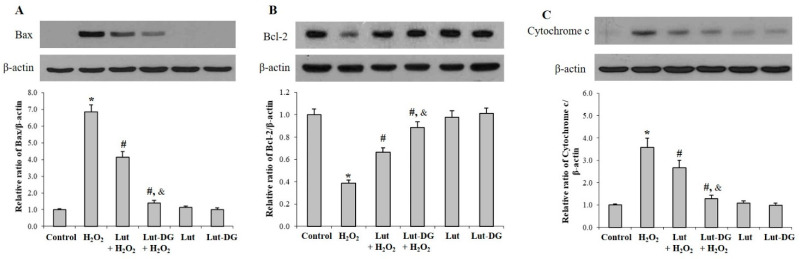
Protective effects of Lut and Lut-DG against oxidative stress occurs through modulation of apoptotic regulatory molecules Bax, Bcl-2 and Cytochrome c. ARPE-19 cells were pre-treated with 1 µM Lut and Lut-DG for 24 h, followed by H_2_O_2_ treatment at 400 µM for 1 h. Protein levels of (**A**) Bax, (**B**) Bcl2 and (**C**) Cytochrome c were evaluated by immunoblotting. β-actin immunodetection was used as a loading control. Representative western blots are shown, with graphs presenting average normalized protein expression. Data is presented as mean ± SD values, *n* = 4; (One-Way ANOVA test), *****
*p* ≤ 0.05 vs. control group, **^#^***p* ≤ 0.05 vs. H_2_O_2_ group, and ^&^ *p* ≤ 0.05 vs. Lut + H_2_O_2_ group).

**Figure 7 ijms-22-04722-f007:**
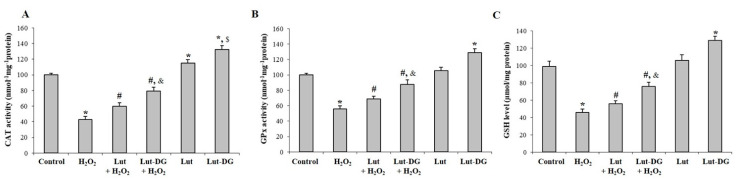
Protective effects of Lut and Lut-DG against oxidative stress occurs through modulation of enzymatic antioxidants (**A**) catalase (CAT) and (**B**) glutathione peroxidase (GPx), and non-enzymatic antioxidant (**C**) glutathione (GSH). ARPE-19 cells were pre-treated with 1 µM of Lut and Lut-DG for 24 h, followed by H_2_O_2_ treatment at 400 µM for 1 h. (**A**) CAT and (**B**) GPx activities and (**C**) GSH levels were assessed by kit assays. Graphs represent the mean ± SD values, *n* = 4; (One-Way ANOVA test), *****
*p* ≤ 0.05 vs. control group, **^#^**
*p* ≤ 0.05 vs. H_2_O_2_ group, ^&^ *p* ≤ 0.05 vs. Lut + H_2_O_2_ group and ^$^
*p* ≤ 0.05 vs. Lut group).

**Figure 8 ijms-22-04722-f008:**
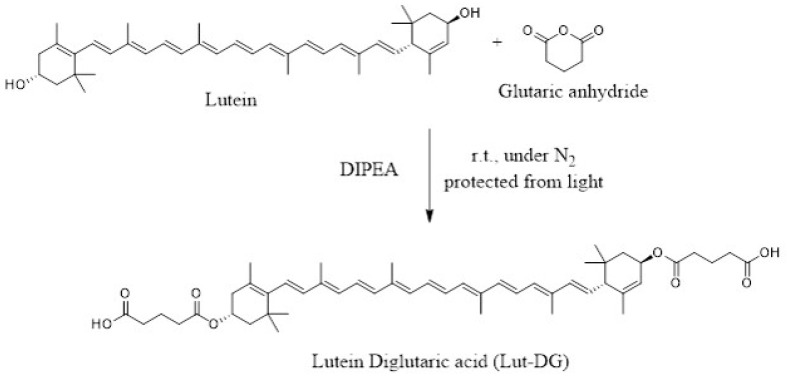
Schematic of the Lutein conjugation reaction with glutaric acid.

**Table 1 ijms-22-04722-t001:** Antibodies used for the analysis of protein expression.

Antibodies	Dilution
Anti-Phospho-ERK1/2 (Cell Signalling, Danvers, USA)	1:1000
Anti-Phospho-p38 (Cell Signalling, Danvers, USA)	1:1000
Anti-Phospho-SAPK/JNK (Cell Signalling, Danvers, USA)	1:1000
Anti-ERK1/2 (Cell Signalling, Danvers, USA)	1:1000
Anti- p38 (Cell Signalling, Danvers, USA)	1:1000
Anti-SAPK/JNK (Cell Signalling, Danvers, USA)	1:1000
Anti-cytochrome C (Cell Signalling, Danvers, USA)	1:1000
Anti-Bax (Cell Signalling, Danvers, USA)	1:1000
Anti-Bcl-2 (Cell Signalling, Danvers, USA)	1:1000
Anti-beta actin (Sigma-Aldrich, Dorset, UK)	1:5000
Secondary horseradish peroxidase (HRP)-conjugated anti-rabbit (Cell Signalling, Hertfordshire, UK)	1:2000
Secondary horseradish peroxidase (HRP)-conjugated anti-mouse (Cell Signalling, Hertfordshire, UK)	1:2000

## Data Availability

Not available.

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
