# Peer review of "Protective Effects of a Lutein Ester Prodrug, Lutein Diglutaric Acid, against H2O2-Induced Oxidative Stress in Human Retinal Pigment Epithelial Cells"

_ijms, 2021, doi:10.3390/ijms22094722_

Round 1
Reviewer 1 Report
In the manuscript “Protective Effects of a Lutein Ester Prodrug, Lutein Diglutaric acid, Against H2O2-induced Oxidative Stress in Human Retinal Pigment Epithelial Cells“, the authors synthesized the lutein derivative, diglutaric acid ester (Lut-DG), and they showed its ability to protect human ARPE-19 cells from the oxidative stress. It turned out that Lut-DG was more effective to protect RPE cells against oxidative stress than Lut, decreasing the H2O2-induced ROS production, modulating phosphorylated MAPKs and downstream molecules: Bax, Bcl-2 and Cytochrome c. Moreover, the authors showed that antioxidants GPx, CAT and GSH were active in the Lut and Lut-DG-treated cells.
In my opinion, the presented manuscript fulfil the criteria of well conducted scientific work, with properly planned experiments to show benefits of the synthesized ester in RPE cells protection against ROS.
Some minor editorial remarks are listed below:
please, check brackets and spaces in the text
row 32 - Lut instead Ltu
in materials and methods, row 424 please, describe applied assay kits
Two topics were neatly omitted here, albeit they are not the main aims of this work and did not diminish results:
- the authors did not show if Lut-DG enter the cell, since its protective action may occurs on the cell surface, scavenging the undesirable ROS, without modulating/inactivating MAPKs pathway and enzymes (radioactive Lut-DG* tracking would be informative here)
- elevated ROS-protective action of Lut-DG in comparison to Lut may results from longer hydrocarbon chain, which better stabilize delocalized electron cloud among entire Lut-DG compound (ROS scavenging ability per one molecule would be informative here)
Author Response
Response to Reviewer’s Comments
Some minor editorial remarks are listed below:
- Please check brackets and spaces in the text
Thank you for your comments. We have checked and amended the brackets and spaces in the revised manuscript as suggested.
- Row 32 - Lut instead of Ltu
We apologize for this mistake. We have changed Ltu to Lut in the abstract section as indicated in red in the revised manuscript (line 35).
- In materials and methods, row 424 please, describe applied assay kits.
We have added the information of the assay kits in the material and methods section as indicated in red in the revised manuscript (lines 428- 456).
Added information of the assay kits (lines 428- 456):
For CAT assays, cell lysate (20 mL) was gently mixed with 100 mL of assay buffer and 100 mL of methanol in each 96-well plate. To initiate the reaction, 20 mL of H2O2 solution (35 mM) was added to the lysate solution. The major end product of this reaction is formaldehyde. Hence, the rate of formaldehyde production from peroxidation reaction can be used to estimate CAT activity. After incubation on a shaker for 20 min, 30 mL of potassium hydroxide (10 M) was added to terminate the reaction. The sample was subsequently incubated with 30 mL of 4-amino-3-hydrazino-5-mercapto-1,2,4-triazole (purpaid) for 10 min. Purpaid is used as a chromogen to specifically interact with the formaldehyde generated from the peroxidation reaction. The 10 mL of potassium periodate (0.5 M) was added into each well to complete the oxidation reaction and develop a purple-color product. After 5 min of incubation, the alteration in color (from colorless to purple) was observed at a wavelength of 540 nm using a SPECTROstar microplate reader. Standard curves with serial dilutions of formaldehyde solution (0 – 85 μM) were used in each experiment to estimate the amount of generated formaldehyde in the sample. The activity of CAT in the sample was further calculated as units per mg of protein.
For GPx assay, cell lysate (20 mL) was mixed with 50 mL of the co-substrate mixture (glutathione reductase and GSH), 50 mL of NADPH and 50 mL of assay buffer in each 96-well plate. To initiate the reaction, 20 mL of cumene hydroperoxide solution was added into each well. After adding cumene hydroperoxide, the absorbance at 340 nm was monitored every 1 min for 10 min with a SPECTROstar microplate reader. The GPx activity in lysate samples was then calculated from a rate of decrease in the absorbance of samples. The GPx activity was expressed as units per mg of protein.
For GSH assay, cell lysate (50 mL) was mixed with the 150 mL of the assay cocktail (GSH MES buffer, GSH co-factor mixture, GSH enzyme mixture and GSH DTNB). The plate was incubated in the dark on an orbital shaker. After 20 min of incubation, the absorbance was observed at a wavelength of 540 nm using a SPECTROstar microplate reader. Standard curves with serial dilutions of GSH (0 – 16 μM) were used in each experiment to estimate the amount of GSH in the sample. The level of GSG in the sample was further calculated as units per mg of protein.
Two topics were neatly omitted here, albeit they are not the main aims of this work and did not diminish results:
- The authors did not show if Lut-DG enters the cell since its protective action may occur on the cell surface, scavenging the undesirable ROS, without modulating/inactivating MAPKs pathway and enzymes (radioactive Lut-DG*tracking would be informative here)
Thank you for this comment. We agree that the protective action may occur both on the cell surface and inside the cell. Lut-DG significantly increased CAT and GPx activities and the GSH level compared to the control cells. The increasing CAT and GPx activities and a GSH level were previously related to the activation of the transcription factor nuclear factor erythroid 2–related factor 2 (Nrf2) located inside the cells [Ref. 50, 51, 69], suggesting that the Lut-DG can enter the cells. However, the possibility of Lut-DG and Lut localization on the surface of the cell using Lut-DG or Lut with radioactive- or fluorescent labeling should be further investigated. We have added this comment in the discussion section as indicated in red in the revised manuscript (lines 292-297),
Added Sentences (lines 292-297):
In addition, the increase of CAT and GPx activities and the GSH level were previously related to the activation of the transcription factor nuclear factor erythroid 2–related factor 2 (Nrf2) located inside the cells [50, 51, 69]. These results suggest that the Lut-DG can enter the cells. However, the possibility of Lut-DG and Lut localization on the surface of the cell using Lut-DG or Lut with radioactive or fluorescent labeling should be further investigated.
- Xu, X. R.; Yu, H. T.; Yang, Y.; Hang, L.; Yang, X. W.; Ding, S. H., Quercetin phospholipid complex significantly protects against oxidative injury in ARPE-19 cells associated with activation of Nrf2 pathway. Eur J Pharmacol 2016, 770, 1-8.
- Huang, S. Y.; Chang, S. F.; Chau, S. F.; Chiu, S. C., The Protective Effect of Hispidin against Hydrogen Peroxide-Induced Oxidative Stress in ARPE-19 Cells via Nrf2 Signaling Pathway. Biomolecules 2019, 9, (8).
- Kamiya, T.; Courtney, M.; Laukkanen, M. O., Redox-Activated Signal Transduction Pathways Mediating Cellular Functions in Inflammation, Differentiation, Degeneration, Transformation, and Death. Oxid Med Cell Longev 2016, 2016, 8479718.
- Elevated ROS-protective action of Lut-DG in comparison to Lut may result from a longer hydrocarbon chain, which better stabilizes the delocalized electron cloud among the entire Lut-DG compound (ROS scavenging ability per one molecule would be informative here)
Thank you for your comment. A previous study showed that the antioxidant activity of lutein is based on the length of the conjugating system on the β‐ionone ring [1]. For instance, lycopene with 11 conjugated double bonds provided higher antioxidant activity than Lutein consisting of 10 conjugated double bonds. However, the addition of glutaric acid to Lut does not affect the length of the conjugating system, suggesting that the increase in ROS-protective action of Lut-DG does not result from an increase of the conjugating system. Ester prodrugs of Lut with various pro-moieties such as palmitate, myristate, linoleate or laureate were more stable than Lut under storage at 25 °C in the dark [56]. In addition, lutein dimyristate was more stable than Lut against heat and UV light conditions than Lut [57]. Therefore, it is likely that the elevated ROS-protective action of Lut-DG compared to Lut may result from a longer hydrocarbon chain, which better stabilizes the delocalized electron cloud of the entire Lut-DG compound. We have added this discussion in the discussion section as indicated in red in the revised manuscript (lines 227-234).
Added Sentences (lines 227-234):
The addition of glutaric acid to Lut does not affect the length of the conjugating system, suggesting that the increase in ROS-protective action of Lut-DG does not result from an increase of the conjugating system. Ester prodrugs of Lut with various pro-moieties such as palmitate, myristate, linoleate or laureate were more stable than Lut under storage at 25 °C in the dark [56]. In addition, lutein dimyristate was more stable than Lut against heat and UV light conditions than Lut [57]. Therefore, it is likely that the elevated ROS-protective action of Lut-DG compared to Lut may result from a longer hydrocarbon chain, which better stabilizes the delocalized electron cloud of the entire Lut-DG compound.
- Jiménez‐Escrig, A.; Jiménez‐Jiménez, I.; Sánchez‐Moreno, C. and Saura‐Calixto, F. Evaluation of free radical scavenging of dietary carotenoids by the stable radical 2,2‐diphenyl‐1‐ Journal of the Science of Food and Argriculture. 2000, 80: 1686-1690
- 56. Gombač, Z.; Črnivec, I.; Skrt, M.; Istenič, K.; Knafelj, A. K.; Pravst, I. and Ulrih, N. P. Stabilisation of Lutein and Lutein Esters with Polyoxyethylene Sorbitan Monooleate, Medium-Chain Triglyceride Oil and Lecithin. Foods. 2021, 10(3), 500.
- 57. Subagio, A.; Wakaki, H. and Morita, N. Stability of Lutein and Its Myristate Esters. Bioscience, Biotechnology, and Biochemistry. 1999, 63(10), 1784-1786.
Reviewer 2 Report
The work by Muangno et al. is insightful to discuss the utility of Lutein diglutaric acid against H2O2-induced oxidative stress. The work reads well, results are well presented (nice and consistent figures), and discussion is aligned. However, before being able to suggest acceptance, I suggest authors to perform the following minor amendments.
First, is H2O2-induced oxidative stress susceptibility a mono-genetic Mendelian trait or rather segregates in a polygenic way. Even in the former scenario, low effect regulators can be presumed. Therefore, I encourage author to check to and cite Nat Rev Genet 2020 21:769-81 during the introduction/discussion sections.
Second, define concrete research questions, goals and hypotheses at the end of the introduction section (L97) in order to enhance readability and coherence.
Third, given potential epistatic factors with the study stress, please refer in the discussion to analogous works (even in other study systems) that have followed a candidate gene approach . Are there known regulators with a strong influence on the studied phenotype? What is the scale of environmental influence?
Fourth, please include a short discussion on how H2O2-induced oxidative stress may become more common in the years to come due to changing habits/environments (e.g. check and cite analogous perspectives in Front Genet 2020 11:656, Front Ecol Evol 2020 8:565708
Fifth, authors must close the Discussion section (L326) by explicitly addressing (1) potential caveats of this study, and (2) pleiotropy with other stresses/major phenotypes (e.g. refer to and cite to the analogous works in other study systems Front Genet 2020 11:564515).
Sixth, please add before the M&M section (L327) an explicit Conclusion and Perspective section in order to add a closure to the writing and bridge major limitations (see above under L326).
Author Response
Response to Reviewer’s Comments
- First is H2O2-induced oxidative stress susceptibility, a mono-genetic Mendelian trait or rather segregates in a polygenic way. Even in the former scenario, low effect regulators can be presumed. Therefore, I encourage the author to check to and cite Nat Rev Genet 2020 21:769-81 during the introduction/discussion sections.
Thank you for your suggestion. We have added the relevant information and the reference of Nat Rev Genet 2020 21:769-81 in the discussion section (Ref 48; line 213) as indicated in red in the revised manuscript.
Added sentences (lines 210-213):
H2O2 was chosen to induce oxidative stress in ARPE-19 cells because ARPE-19 cells have a high metabolic rate and exist in an environment that is abundant in endogenous ROS, such as O2-, H2O2 and OH-. Long-term accumulation of oxidative damage leads to dysfunction of RPE cells and increases their susceptibility to oxidative stress [47, 48].
- Blasiak, J.; Barszczewska, G.; Gralewska, P.; Kaarniranta, K., Oxidative stress induces mitochondrial dysfunction and autophagy in ARPE-19 cells. Acta Ophthalmologica 2019, 97, (S263).
- Barghi, N.; Hermisson, J.; Schlötterer, C., Polygenic adaptation: a unifying framework to understand positive selection. Nature Reviews Genetics 2020, 21, (12), 769-781.
- Second, define concrete research questions, goals and hypotheses at the end of the introduction section (L97) in order to enhance readability and coherence.
Thank you for your suggestion. We have modified the last paragraph of the introduction section to include the research questions, goals and hypothesis of the study as indicated in red in the revised manuscript (lines 102-112).
Added sentences (lines 102-112):
The prodrug approach has been shown to enhance pharmacological properties by improving physico-chemical and biopharmaceutical properties such as aqueous solubility, stability, bioavailability and biological activities [36-42]. We have previously demonstrated that curcumin digluatric acid, an ester prodrug of curcumin conjugated with diglutaric acid, improved the biological activities of curcumin both in vitro and in vivo [41, 42], suggesting that glutaric acid could serve as a promoiety of bioactive molecules. The conjugation of Lut with diglutaric acid via an ester bond can possibly increase the pharmacological and biological activities of Lut. Therefore, in this study, we synthesized a novel ester prodrug of Lut, namely lutein diglutaric acid (Lut-DG). Lut-DG and Lut were evaluated on their protective effect against oxidative stress induced by H2O2 in RPE cells. The underlying molecular mechanisms by which these drugs exert their effects were also explored.
- Rautio, J.; Kumpulainen, H.; Heimbach, T.; Oliyai, R.; Oh, D.; Järvinen, T.; Savolainen, J., Prodrugs: design and clinical applications. Nature Reviews Drug Discovery 2008, 7, 255.
- Abet, V.; Filace, F.; Recio, J.; Alvarez-Builla, J.; Burgos, C., Prodrug approach: An overview of recent cases. European Journal of Medicinal Chemistry 2017, 127, 810-827.
- Ratnatilaka Na Bhuket, P.; El-Magboub, A.; Haworth, I. S.; Rojsitthisak, P., Enhancement of Curcumin Bioavailability Via the Prodrug Approach: Challenges and Prospects. European Journal of Drug Metabolism and Pharmacokinetics 2017, 42, (3), 341-353.
- Muangnoi, C.; Ratnatilaka Na Bhuket, P.; Jithavech, P.; Supasena, W.; Paraoan, L.; Patumraj, S.; Rojsitthisak, P., Curcumin diethyl disuccinate, a prodrug of curcumin, enhances anti-proliferative effect of curcumin against HepG2 cells via apoptosis induction. Scientific Reports 2019, 9, (1), 11718.
- Muangnoi, C.; Ratnatilaka Na Bhuket, P.; Jithavech, P.; Wichitnithad, W.; Srikun, O.; Nerungsi, C.; Patumraj, S.; Rojsitthisak, P., Scale-Up Synthesis and In Vivo Anti-Tumor Activity of Curcumin Diethyl Disuccinate, an Ester Prodrug of Curcumin, in HepG2-Xenograft Mice. Pharmaceutics 2019, 11, (8).
- Phumsuay, R.; Muangnoi, C.; Dasuni Wasana, P. W.; Hasriadi; Vajragupta, O.; Rojsitthisak, P.; Towiwat, P., Molecular Insight into the Anti-Inflammatory Effects of the Curcumin Ester Prodrug Curcumin Diglutaric Acid In Vitro and In Vivo. International Journal of Molecular Sciences 2020, 21, (16).
- Muangnoi, C.; Jithavech, P.; Ratnatilaka Na Bhuket, P.; Supasena, W.; Wichitnithad, W.; Towiwat, P.; Niwattisaiwong, N.; Haworth, I. S.; Rojsitthisak, P., A curcumin-diglutaric acid conjugated prodrug with improved water solubility and antinociceptive properties compared to curcumin. Biosci Biotechnol Biochem 2018, 82, (8), 1301-1308.
- Third, given potential epistatic factors with the study stress, please refer in the discussion to analogous works (even in other study systems) that have followed a candidate gene approach. Are there known regulators with a strong influence on the studied phenotype? What is the scale of environmental influence?
Thank you very much for raising this exciting issue. We lack knowledge in the epistatic factors with oxidative stress studies. We, therefore, apologize that we cannot discuss authoritatively the above matter and believe that this does not affect the interpretation or discussion of the results of this study.
- Fourth, please include a short discussion on how H2O2-induced oxidative stress may become more common in the years to come due to changing habits/environments (e.g., check and cite analogous perspectives in Front Genet 2020 11:656, Front Ecol Evol 2020 8:565708
Thank you for your suggestion. We have added the discussion on how H2O2- induced oxidative stress in the discussion section as indicated in red in the revised manuscript (lines 210-217).
Added sentences (lines 210-217):
H2O2 was chosen to induce oxidative stress in ARPE-19 cells because ARPE-19 cells have a high metabolic rate and exist in an environment that is abundant in endogenous ROS, such as O2-, H2O2 and OH-. Long-term accumulation of oxidative damage leads to dysfunction of RPE cells and increases their susceptibility to oxidative stress [47, 48]. H2O2 is one of the most common oxidants used in the cellular oxidative stress models for ARPE-19 cells [49-51]. The increase of the intracellular H2O2 level in response to various pro-oxidants can further induce excessive ROS production in the cells which leads to the RPE cells dysfunction and death from oxidative.
- Blasiak, J.; Barszczewska, G.; Gralewska, P.; Kaarniranta, K., Oxidative stress induces mitochondrial dysfunction and autophagy in ARPE-19 cells. Acta Ophthalmologica 2019, 97, (S263).
- Barghi, N.; Hermisson, J.; Schlötterer, C., Polygenic adaptation: a unifying framework to understand positive selection. Nature Reviews Genetics 2020, 21, (12), 769-781.
- Kaczara, P.; Sarna, T.; Burke, J. M., Dynamics of H2O2 availability to ARPE-19 cultures in models of oxidative stress. Free Radic Biol Med 2010, 48, (8), 1064-70.
- Xu, X. R.; Yu, H. T.; Yang, Y.; Hang, L.; Yang, X. W.; Ding, S. H., Quercetin phospholipid complex significantly protects against oxidative injury in ARPE-19 cells associated with activation of Nrf2 pathway. Eur J Pharmacol 2016, 770, 1-8.
- Huang, S. Y.; Chang, S. F.; Chau, S. F.; Chiu, S. C., The Protective Effect of Hispidin against Hydrogen Peroxide-Induced Oxidative Stress in ARPE-19 Cells via Nrf2 Signaling Pathway. Biomolecules 2019, 9, (8).
- Fifth, authors must close the Discussion section (L326) by explicitly addressing (1) potential caveats of this study and (2) pleiotropy with other stresses/major phenotypes (e.g., refer to and cite to the analogous works in other study systems Front Genet 2020 11:564515).
Thank you for your comments. This study investigated the protective effect and mechanism of Lut-DG on H2O2-induced oxidative stress in ARPE-19 cells. The oxidative stress in ARPE-19 cells plays a crucial role in cell dysfunction and death, which leads to AMD progression and development. Our results indicate that Lut-DG is a promising prodrug of Lut and has the potential to be extensively investigated as a therapeutic agent or an adjuvant for the treatment of AMD in additional preclinical and clinical studies. Our results showing the protective benefit of Lut-DG on ARPE-19 cells using H2O2 as an oxidative stress inducer may potentially be extended to other oxidative stress inducers such as light, smoke and pollution. It has been known that oxidative stress inducer such as H2O2, light and smoke have indicated in pleiotropy resulting in the genetic level. Further investigation of the Lut-DG action should be performed on their effects at the genetic level.
We have added the some sentences in the discussion section as indicated in red in the revised manuscript (lines 306-311).
Added sentences (lines 306-311):
This study investigated the protective effect and mechanism of Lut-DG on H2O2-induced oxidative stress in ARPE-19 cells. The oxidative stress in ARPE-19 cells plays a crucial role in cell dysfunction and death, which leads to AMD progression and development. Our results indicate that Lut-DG is a promising prodrug of Lut and has the potential to be extensively investigated as a therapeutic agent or an adjuvant for the treatment of AMD in additional preclinical and clinical studies.
- Sixth, please add before the M&M section (L327) an explicit Conclusion and Perspective section in order to add a closure to the writing and bridge major limitations (see above under L326).
Thank you for your suggestion. We have added the conclusion section before the materials and methods section as indicated in red in the revised manuscript (lines 313-326).
Added Conclusion (lines 313-326):
In summary, the study provides evidence that the impact/effect of Lut and Lut-DG on the apoptosis-related proteins is oxidative stress-dependent in human ARPE-19 cells exposed to H2O2. Therefore, the H2O2-induced oxidative stress in human ARPE-19 cells is applicable as a model for evaluating the ability of lutein and its prodrug, lutein diglutaric acid (Lut-DG), to protect these cells from oxidative stress. We demonstrate that Lut-DG is a more potent protective antioxidant than Lut against oxidative damage to RPE cells by suppressing ROS levels and protecting from cell death. The molecular mechanisms by which Lut-DG showed its effects were through specific modulation of apoptotic signaling (p38, ERK1/2 and SAPK/JNK) and downstream proteins such as Bax, Bcl2 and cytochrome c as well as enhancement of activities of enzymatic antioxidants CAT and GPx and the level of non-enzymatic antioxidant GSH. The results provide evidence for Lut-DG as a promising therapeutic agent for the treatment of AMD.